# Comparative study of thoracolumbar fascia changes in weightlifters with and without low back pain

Emma Feng Ming Zhou[1], Carmelo Pirri[2], Xiaoxiao Zhao[2], Juhua Peng[3], Tao Wen[3], Jianhui Fang[3], Chufeng Zhou[3], Siu Ngor Fu[1]*, Carla Stecco[2]

1 Department of Rehabilitation Sciences, The Hong Kong Polytechnic University, Hung Hom, Hong Kong, 2 Department of Neurosciences, Institute of Human Anatomy, University of Padova, Padova, Italy, 3 Department of Rehabilitation, Guangdong Sports Hospital, Guangzhou, Guangdong, China

* amy.fu@polyu.edu.hk

## Abstract

### Background

Thoracolumbar fascia (TLF) is thought to be linked to low back pain (LBP). This cross-sectional study aimed to investigate the difference in TLF thickness and stiffness between elite weightlifters with and without chronic LBP.

### Methods

Forty-six elite weightlifters (aged 16–26 years; 23 with chronic bilateral LBP) were recruited. TLF thickness was measured in all participants, while stiffness was assessed in 17 of them (aged 19–25 years; 9 with chronic bilateral LBP). Investigations were conducted at the L3 level, about 2 cm lateral to midline, using ultrasound. Bright mode and shear wave elastography (SWE) mode were employed to measure the thickness and Young's modulus (as an indicator of stiffness) of TLF, respectively. Analyses of covariates (ANCOVAs) were employed to compare the differences in TLF thickness and stiffness between LBP and non-LBP groups, controlling for pre-determined confounding factors. Repeated ANCOVAs were performed to investigate the side-to-side differences in TLF thickness and stiffness in weightlifters with and without LBP. The significance level was set as $p \leq 0.05$.

### Results

The mean thickness and stiffness of TLF were $1.63 \pm 0.38$ mm and $47.77 \pm 13.26$ kPa on the dominant, and $1.88 \pm 0.60$ mm and $48.09 \pm 12.62$ kPa on the non-dominant sides respectively. The stiffness of the TLF on the dominant side was 42.4% higher in the LBP group compared to the non-LBP control (mean difference (MD) $=16.55$ kPa $>$ MDC$_{95}$, $p = 0.005$, Cohen's d $= 1.58$). No significant differences were detected in TLF thickness ($p > 0.05$). Additionally, LBP was found to be a factor influencing

**Data availability statement:** The datasets generated and analyzed during this study contain individual-level information from elite athletes and cannot be publicly shared due to confidentiality restrictions under the institutional collaboration agreement between The Hong Kong Polytechnic University and the Ersha Sports Training Center of Guangdong. Public deposition of these data may risk indirect participant re-identification and would violate the terms of the collaboration agreement. Anonymized data underlying the main findings of this study may be accessed upon reasonable request. Requests will be evaluated to ensure compliance with institutional regulations and data confidentiality. Please direct inquiries to one of the following contacts: Institutional Review Board of The Hong Kong Polytechnic University (Chair; non-author): Email: marco.pang@polyu.edu.hk Administrative Office of Ersha Sports Training Center of Guangdong: Email: gdesbgs@163.com.

**Funding:** The author(s) received no specific funding for this work.

**Competing interests:** The authors have declared that no competing interests exist.

**Abbreviations:** ANCOVA, analyses of covariate; BMI, body mass index; B-mode, bright-mode; CI, confidence interval; dom/ndom, the dominant side/the non-dominant side; ICC, intraclass correlation coefficient; LBP, low back pain; $MDC_{95}$, minimal detectable change with 95% confidence; MD, mean difference; NLBP, non-low back pain; NPRS, Numeric Pain Rating Scale; Q-box, quantification box; ROI, region of interest; SEM, standard error of measurement; SWE, shear wave elastography; TLF, thoracolumbar fascia; L3, the third lumbar vertebra.

the side-to-side differences in stiffness but not in thickness. Specifically, the thickness of the non-dominant side was 15.3% higher than the dominant side (MD = 0.25 mm > $MDC_{95}$, $p < 0.001$, Cohen's d = 0.63), which was not detected in stiffness.

## Conclusion

Chronic LBP in elite weightlifters was associated with significantly higher TLF stiffness but unchanged thickness, suggesting stiffness is a more informative indicator of TLF health than thickness. Addressing stiffness in prevention and rehabilitation programs may improve weightlifters' performance and career longevity.

## Background

Low back pain (LBP) is a major concern among weightlifters, with a reported one-year prevalence of 54–85% [1,2]. This high incidence of LBP not only compromises their performance but also threatens their careers, leading to persistent episodes throughout their lifetime [3]. Weightlifting subjects the lower back to considerable stress, typically exposing it to an average compressive load exceeding 17,192N [4,5]. Such intense loading demands robust lumbar stabilization and efficient force transmission to mitigate the risk of LBP. The thoracolumbar fascia (TLF), situated in the lower back and connected to vertebrae, ligaments, and multiple muscles, plays a pivotal biomechanical role in maintaining spinal stability and transmitting force between the trunk and lower limbs [6,7], and is thought to be a crucial factor in LBP.

TLF is rich in nociceptors [8,9], and research has reported that TLF is more sensitive to pain stimuli than muscles [10]. These findings have led researchers to consider the TLF itself as an important source of LBP. However, how TLF dysfunction relates to LBP still requires further exploration. Langevin et al. were the first to report increased lumbar connective tissue thickness in individuals with LBP using ultrasound, even though this measurement was of the perimuscular connective tissue, without distinguishing the thoracolumbar fascia and the epimysium of the erector spine muscles [11]. More recently, Pirri et al. using clearly defined measurement protocols for the TLF, found that individuals with chronic LBP had increased TLF thickness compared to healthy individuals [12]. Contrarily, another study on amateur athletes reported no difference in thickness but observed higher disorganization of TLF in the LBP group [13]. Additionally, Langevin et al. in 2011 using ultrasound shear strain, the first introduced ultrasound elastography technique, evaluated the shear mobility of the fascia and revealed a 20% reduction in shear mobility (stiffness) in the LBP group [14]. These prior studies suggest changes in both morphology (thickness) and mechanical properties (stiffness) in the context of chronic LBP; however, these changes may vary between athletes and the general population.

It is known that the molecular and cellular activity, collagen turnover, and subsequently morphology and mechanical properties of connective tissue can change in response to mechanical loading [15]. Sports-specific adaptations may be present in athletes due to the unique demands of their training. Weightlifters, who experience

repetitive extreme loading during prolonged training, are significantly different from the general population in this regard [5]. As a result, the TLF of weightlifters may undergo adaptive remodeling, leading to different patterns of thickness and stiffness adaptation compared to the general population.

Given the above, our study focuses on elite weightlifters who endure long-term extreme loading on the lower back. Firstly, it aims to investigate the difference in TLF thickness and stiffness between elite weightlifters with and without chronic LBP. Secondly, we aim to explore the side-to-side difference (dominant and non-dominant sides) in thickness and stiffness in weightlifters with and without LBP. Considering that ultrasound shear strain investigation requires externally applied stress, which is not quantifiable [16], and that the currently well-developed ultrasound shear wave elastography (SWE) is widely applied in research and clinical practice as an objective and reliable method to quantify tissue stiffness [16,17], we would use ultrasound SWE in the present study to quantify TLF stiffness. We hypothesize that both thickness and stiffness would be increased in athletes with chronic LBP compared to asymptomatic controls; the thickness and stiffness would be different between the dominant and non-dominant sides.

## Methods

### Participants recruitment

This is a cross-sectional study. Participants were consecutively recruited from the weightlifting team at the Sports Training Center between May 5 and June 30, 2023, using a convenience sampling approach. To meet the inclusion criteria in the chronic LBP group, weightlifters needed to be between 16 and 30 years old [18], have achieved national level [19], suffer from chronic LBP characterized by non-specific pain or discomfort between the lower rib margins and the buttock creases, present during both daily life and athletic training, persisting or fluctuating for more than 3 months [20–22], with an average pain intensity of at least 3 out of 10 on the Numeric Pain Rating Scale (NPRS) over the past 7 days [23]. Exclusion criteria included pathological LBP conditions (such as malignancy, epidural abscess, spinal canal stenosis, spondyloarthropathy, radiculopathy, cauda equina syndrome, compression fracture), neurological disorders, malignant tumors, severe organ diseases, previous spinal surgery, or other conditions affecting current pain severity or daily training [20]. These conditions were excluded based on medical records and evaluations conducted by team physicians. Weightlifters who had been free of LBP for the past 3 months were selected as non-LBP controls [24].

This observational study was approved by the Institutional Review Board of the Hong Kong Polytechnic University (Reference Number: HSEARS20220527005). Prior to data collection, informed written consent was obtained from each participant, as well as from the guardians of participants under 18. The study includes two parts: the first part was for TLF thickness investigation, and the second part was for SWE (stiffness) investigation. Intra-rater reliability of TLF thickness and SWE measurements was established before the main study. Demographic information, including age, sex, body mass index (BMI), fat percentage, years of training, and dominant side (the leading leg during lifting), was collected.

### Ultrasound measurements

Ultrasonic investigations were performed by a diagnostic medical sonographer with 20 years of experience, who was blinded to the demographic information and LBP history of the participants.

The Supersonic Imagine's Aixplorer® (France) system was used to quantify the thickness and stiffness of TLF. Athletes were positioned prone with arms relaxed at their sides. Using Bright-mode (B-mode), a linear transducer (4–15 MHz) was placed with minimal pressure on the spinous process of the third lumbar vertebra (L3) and then swept approximately 2 cm laterally to locate the L2/3 and L3/4 facet joint, which served as references for TLF [11,12]. The probe was held parallel to the spinous process and perpendicular to the skin, clearly defining the TLF in this region (Fig 1A and 1B). The images with an optimum view of the fascia on both sides were captured for later offline thickness measurements.

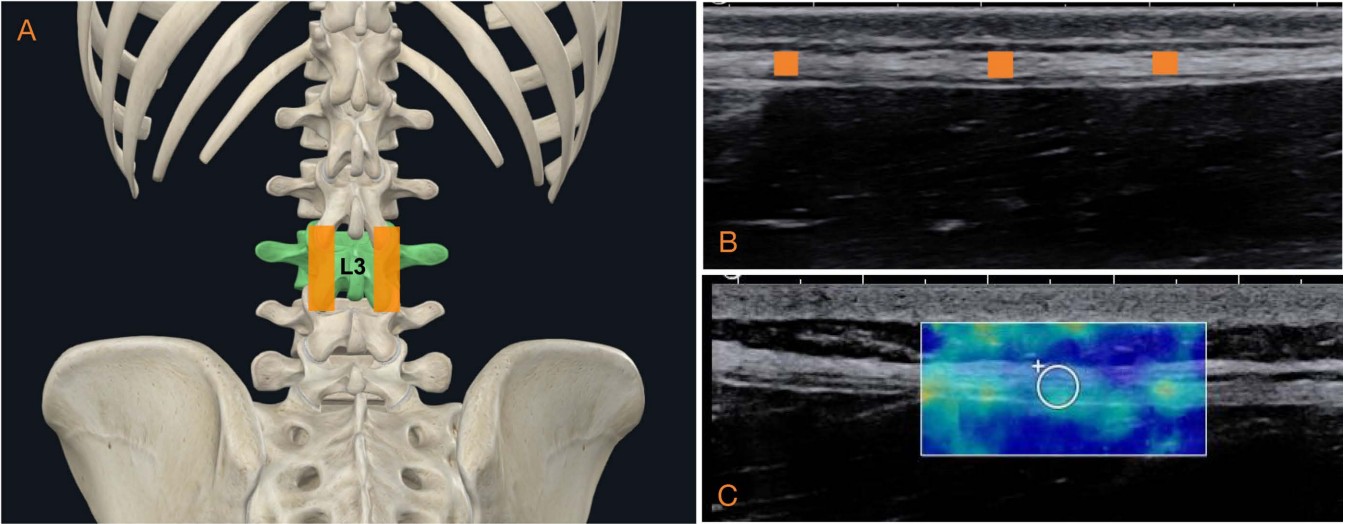

**Fig 1. The measurement of TLF with ultrasound.** 1A: location of transducer; 1B: TLF thickness measurement under B-mode; 1C: TLF stiffness measurement under shear wave elastography mode. Abbreviations: TLF = thoracolumbar fascia; B-mode = bright mode.

For elastography imaging, the location with B-mode was the same as the thickness measurement, then the system was subsequently switched to SWE mode, and a rectangular region of interest (ROI) was positioned over the identified TLF area. The ROI was color-filled to represent Young's modulus magnitude. Three images with homogenous and well-filled ROIs were captured (Fig 1C). To avoid the influence of breathing, images are captured at the end of tidal expiration [25].

Offline analyses were conducted after capturing all images to prevent recall bias. TLF thickness was measured using Image J software (NIH, USA). Following Pirri et al's [12] protocol for defining and measuring TLF layers under ultrasound, each image was divided into three equidistant regions. Within each region, three points with the best visibility were measured and averaged to account for potential thickness variations (Fig 1B). For SWE analyses, a circled quantification box (Q-box) was drawn within each ROI, maximizing size without overlapping and maintaining a standard deviation within 20% (Fig 2C). The mean Young's modulus (in kPa) within the Q-box was averaged from three recorded images.

## Statistical analysis

Statistical analyses were conducted using IBM SPSS version 26.0 (IBM Corp., Armonk, NY, USA). The normality of the data was assessed with the Shapiro-Wilk test. Intra-rater consistency was reported as intraclass correlation coefficient model 3 ($ICC_{3,1}$, single measurement) with corresponding 95% confidence intervals. Standard error of measurement (SEM) and minimal detectable change with 95% confidence ($MDC_{95}$) were calculated using the formulas, $SEM = standard\ deviation * \sqrt{(1-ICC)}$ and $MDC_{95} = 1.96 * SEM * \sqrt{2}$, respectively [26]. Analyses of covariates (ANCOVAs) were employed to compare the differences of TLF thickness and stiffness between LBP and non-LBP groups. Post-hoc pairwise comparisons were performed when $p \leq 0.05$. Repeated ANOCVAs and subsequent pair t-tests were used to investigate the side-to-side difference in TLF thickness and stiffness with and without LBP. Demographic variables with significant between-group differences ($p < 0.1$), including age, BMI, and years of training, were entered as covariates in the analysis. The significance level was set as $p \leq 0.05$. In addition, the statistical power of the main ANCOVA analyses was calculated and reported to verify that the sample size was sufficient to detect meaningful between-group differences.

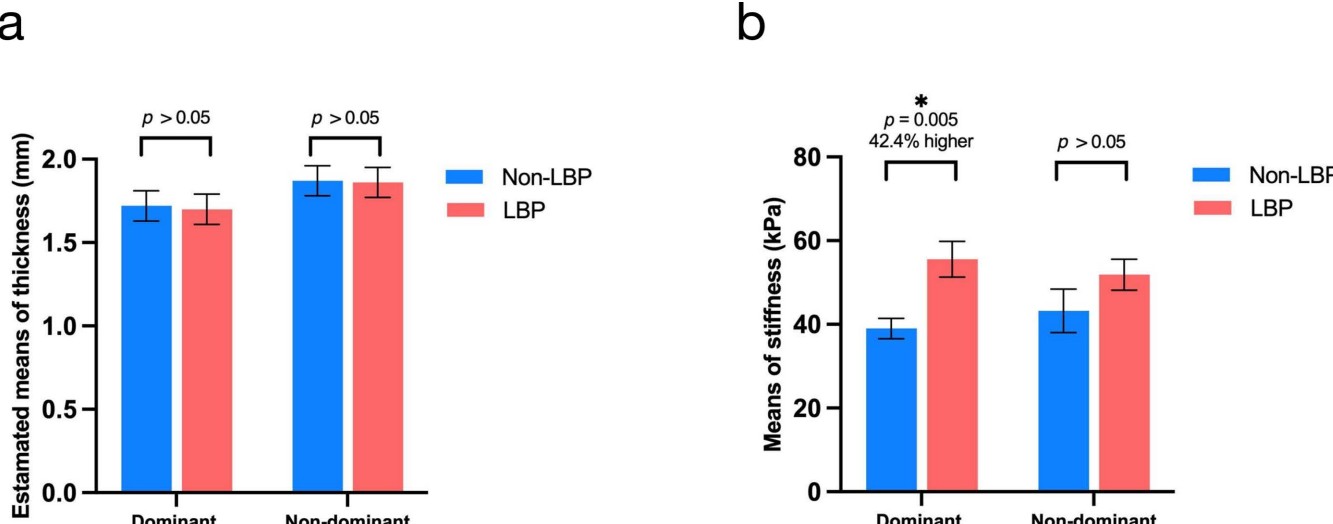

**Fig 2. Comparison of TLF thickness and stiffness in weightlifters with and without chronic LBP.** a: comparison of thickness using ANCOVAs with age, BMI, and years of training as covariates; b: comparison of stiffness with independent t-tests. Abbreviations: LBP = low back pain; TLF = thoracolumbar fascia; BMI = body mass index. *: $p < 0.05$.

## Inclusivity in global research

Additional information regarding the ethical, cultural, and scientific considerations specific to inclusivity in global research is included in the Supporting Information (S1 Checklist).

## Results

Given the inherent limitations in the number of elite athletes available and the constraints imposed by their training schedules, a total of 48 elite weightlifters were recruited (aged 16–26 years, 25 with chronic LBP, 2 with unilateral LBP) were recruited for the thickness study, and 18 elite weightlifters (aged 19–25 years, 10 with chronic LBP, 1 with unilateral pain) were recruited for the stiffness study. Considering the potential difference between bilateral and unilateral LBP, only those athletes with bilateral LBP were included for analysis in the LBP group. Table 1 shows their demographic information. In the cohort for thickness measurement, the LBP group had a significantly higher height, weight, and BMI. However, no differences were detected between the two groups in the SWE investigation cohort.

Similar to prior studies [12,25,27], excellent and good-to-excellent intra-rater test-retest reliability on 16 participants was demonstrated in quantifying the thickness ($ICC_{3,1} = 0.977$–$0.984$, $MDC_{95} = 0.14$–$0.15$ mm) and stiffness (Young's modulus) of TLF ($ICC_{3,1} = 0.861$–$0.923$, $MDC_{95} = 10.98$–$12.49$ kPa), respectively (S1 Table).

### Comparison of thickness and stiffness between LBP and non-LBP group

Table 2 shows the mean thickness and stiffness (Young's modulus) of dominant and non-dominant sides. ANCOVA analysis showed no notable differences in thickness between LBP and non-LBP groups when age, BMI, and years of training were included as covariates. However, the stiffness on the dominant side was 42.4% higher in the LBP group compared to the non-LBP control (mean difference (MD) = 16.55 kPa > $MDC_{95}$, $p = 0.005$, Cohen's d = 1.58). No detectable difference was found on the non-dominant side (MD = 8.64 kPa, $p = 0.183$) (Fig 2).

**Table 1. Demographics of included participants (mean ± SD).**

| Thickness | Total (*n* = 46) | NLBP (*n* = 23) | LBP (*n* = 23) | *p* value |
|---|---|---|---|---|
| Age (y) | 20.52 ± 2.68 | 19.78 ± 2.37 | 21.26 ± 2.80 | 0.060[a] |
| Height (m) | 1.63 ± 0.09 | 1.59 ± 0.09 | 1.66 ± 0.09 | 0.008* |
| Weight (kg) | 68.45 ± 14.23 | 62.25 ± 11.53 | 74.65 ± 14.18 | 0.002* |
| BMI (kg/m²) | 25.64 ± 2.78 | 24.40 ± 1.97 | 26.88 ± 2.95 | 0.002* |
| Sex (male/female) | 20/26 | 7/16 | 13/10 | 0.136 |
| Training years (y) | 8.65 ± 2.96 | 7.83 ± 2.41 | 9.48 ± 3.27 | 0.058[a] |
| Fat (%) | 16.74 ± 4.42 | 16.74 ± 3.87 | 16.73 ± 5.02 | 0.993 |
| SWE | Total (*n* = 17) | NLBP (*n* = 8) | LBP (*n* = 9) | *p* value |
| Age (y) | 21.35 ± 1.66 | 21.63 ± 1.77 | 21.11 ± 1.62 | 0.540 |
| Height (m) | 1.65 ± 0.09 | 1.66 ± 0.11 | 1.64 ± 0.06 | 0.583 |
| Weight (kg) | 71.98 ± 13.87 | 74.13 ± 16.76 | 70.08 ± 11.40 | 0.565 |
| BMI (kg/m²) | 26.24 ± 2.32 | 26.53 ± 2.46 | 25.99 ± 2.30 | 0.647 |
| Sex (male/female) | 9/8 | 4/4 | 5/4 | 0.819 |
| Training years (y) | 9.41 ± 3.81 | 9.75 ± 2.38 | 9.11 ± 1.97 | 0.639 |
| Fat (%) | 16.39 ± 3.81 | 16.83 ± 3.93 | 16.00 ± 3.90 | 0.671 |

Abbreviations: SD = standard deviation; NLBP = non-low back pain; LBP = low back pain; BMI = body mass index; SWE = shear wave elastography. [a]: *p* < 0.1; *: *p* < 0.05.

**Table 2. Thickness and stiffness of TLF in weightlifters with and without LBP (mean ± SD).**

| | Total | NLBP | LBP | F | *p* value | *Observed Power* |
|---|---|---|---|---|---|---|
| Thickness_dom (mm)[a] | 1.63 ± 0.38 | 1.60 ± 0.37 | 1.83 ± 0.55 | 0.02 | 0.881 | 0.05 |
| Thickness_ndom (mm)[a] | 1.88 ± 0.60 | 1.73 ± 0.46 | 1.99 ± 0.52 | 0.01 | 0.934 | 0.05 |
| Stiffness_dom (kPa)[b] | 47.77 ± 13.26 | 39.01 ± 6.86 | 55.56 ± 12.86 | | 0.005* | |
| Stiffness_ndom (kPa)[b] | 48.09 ± 12.62 | 43.23 ± 13.67 | 51.87 ± 11.04 | | 0.183 | |

Abbreviations: TLF = thoracolumbar fascia; SD = standard deviation; NLBP = non-low back pain; LBP = low back pain; Thickness_dom/ndom = thickness of the dominant/non-dominant side; Stiffness_dom/ndom = stiffness of the dominant/non-dominant side. [a]: ANCOVAs with age, BMI, and years of training as covariates; [b]: Independent t-tests; *: *p* < 0.05.

## Side-to-side differences in thickness and stiffness

Repeated ANCOVA demonstrated that LBP was not a significant factor influencing side-to-side differences in thickness (F = 0.03, *p* = 0.876) taking age, BMI, and years of training as confounding factors. Then paired t-tests were applied to compare the differences between both sides. The non-dominant side was 15.3% thicker than the dominant side (MD = 0.25 mm > MDC$_{95}$, *p* < 0.001, Cohen's d = 0.63).

However, repeated ANOVA showed that LBP was a significant factor influencing side-to-side differences in stiffness (F = 5.26, *p* = 0.038, Observed power = 0.57). Paired t-tests for bilateral comparison in non-LBP and LBP groups were conducted respectively, and no significant differences were found (Fig 3).

## Discussion

In this study, we observed a 42% higher TLF stiffness in elite weightlifters with LBP than those without. This increase in stiffness was noted on the dominant side but not on the non-dominant side. However, there were no significant differences in TLF thickness between the two groups. Moreover, LBP was found to be a factor influencing the side-to-side differences

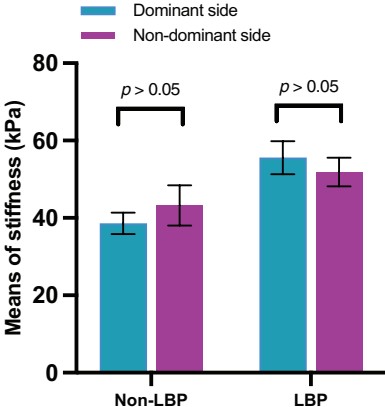

**Fig 3. The stiffness of bilateral TLF in weightlifters with and without LBP.** Comparison of TLF stiffness between dominant and non-dominant sides in the non-LBP group and in the LBP group, respectively. Abbreviations: LBP = low back pain.

in stiffness but not in thickness. Specifically, the TLF on the non-dominant side was thicker than that on the dominant side, a difference that was not observed in stiffness.

## Higher TLF stiffness in weightlifters with chronic LBP

To our knowledge, this is the first study to investigate TLF stiffness using SWE and to compare differences between LBP and non-LBP conditions. Our findings align with an earlier study by Langevin et al., [14] which reported stiffer TLF in the general chronic LBP population measured with ultrasound shear strain.

Fascial stiffness may arise from multiple interacting mechanisms. Chronic mechanical overload and microtrauma can trigger extracellular matrix (ECM) remodeling, characterized by increased collagen deposition, altered fiber orientation, and excessive cross-linking, which reduce elasticity and the gliding ability of tissue layers [15,28,29]. In addition, chronic inflammation can promote fibrosis through cytokine-mediated activation of fibroblasts and myofibroblasts, while hyaluronan accumulation increases connective tissue viscosity and adhesion between fascial layers, further elevating passive stiffness [30–34]. These structural and biochemical changes impair the tissue's ability to deform and transmit loads efficiently, predisposing the fascia to become stiffer and less resilient.

The stiffer TLF could be a cause of LBP. TLF comprises a network of collagen fibers that form a complex arrangement of multiple fascia layers. The orientation of these fibers in each layer varies and can glide over each other, allowing and facilitating the collagen fibers to adapt and remodel according to functional demands [35,36]. External factors, such as unoptimized technique and improper compensation habits during long-term high-loading training, could impose unexpected stress or overuse on the lower back [37]. In response to these stresses, the extracellular matrix may remodel along unexpected directions to make the tissue more damage-resistant and to ensure optimal force transmission [15,28,29]. However, this adaptive process may lead to disorganization of collagen fibers, thereby increasing stiffness [13]. Ultrasonography findings also indicated a loss of organization and anisotropy in the LBP population [12]. Additionally, overuse induced both morphological and biochemical changes, including altered collagen typing and fibril size, and impaired collagen degradation activity, thereby increasing stiffness [15]. Increased stiffness indicates the loss of elasticity, impairing the capacity of energy storage, absorption, and transfer [5]. It also results in compromised mechanical properties that govern the response to load, significantly lowering the load tolerance threshold, and thereby increasing the risk of LBP of weightlifters [37]. In addition, stiffer TLF may result in restricted joint and muscle movement and function, compromise proprioception, and altered innervation and circulation, potentially inducing LBP [9,38,39].

Conversely, persistent or recurrent LBP itself may further exacerbate fascial stiffness. Chronic pain has been shown to induce sustained low-level muscle activation and altered motor control [40,41], which impose continuous mechanical tension on the fascia and facilitate extracellular matrix remodeling over time [14]. In addition, local inflammatory processes may promote collagen cross-linking and structural reorganization within the fascia, further reinforcing tissue stiffness. Tissue fibrosis has also been observed in chronic LBP conditions [41], and the acidic environment associated with chronic inflammation can enhance hyaluronan accumulation, increasing connective tissue viscosity and reducing interlayer gliding [30–34]. Collectively, these alterations may contribute to a self-perpetuating cycle in which stiffness exacerbates LBP and, in turn, LBP promotes further stiffness.

Furthermore, it remains unclear whether fascial stiffness fully normalizes once LBP resolves. Although participants in the non-LBP group were pain-free for at least three months [20–22], residual fascial or neuromuscular adaptations from prior loading or pain episodes may persist. Structural remodeling, such as collagen cross-linking or fibrosis, may reverse more slowly than pain symptoms, potentially leaving subtle mechanical alterations even after symptom resolution. Longitudinal or interventional studies tracking stiffness changes before and after recovery would help clarify the reversibility and temporal dynamics of these adaptations.

While this study focused on the structural and mechanical characteristics of the thoracolumbar fascia, the underlying nociceptive mechanisms were not examined. Given that fascia is richly innervated and may contribute to pain sensitization when its mechanical properties are altered [42], future studies integrating both mechanical and nociceptive assessments are warranted to clarify this relationship more comprehensively.

Notably, the higher stiffness in the LBP group was only found on the dominant side (the leading leg during lifting). This may relate to the loading from sports-specific movement patterns [15]. During weightlifting, the dominant leg is positioned in front, lengthening the TLF on the same side, while the non-dominant leg is positioned in the back, keeping the ipsilateral TLF in a shortened condition. The dominant side may have a higher reliance/demand on TLF elasticity due to the specific movement patterns in weightlifting. Therefore, a decrease in elasticity (increased stiffness) on the dominant side is more likely to lead to functional impairments and subsequently contribute to the development of LBP.

**TLF thickness adaption in chronic LBP**

The present study on elite weightlifters found TLF thickness comparable to that reported in the general population [12], with a mean thickness of 1.75 ± 0.85 mm in the non-LBP group and 2.11 ± 0.65 mm in the LBP group, as well as in amateur athletes, which reported a mean thickness of 1.60 ± 0.40 mm in the non-LBP group and 1.70 ± 0.40 mm in the LBP group [13]. This may suggest that TLF thickness does not notably increase under long-term high-weight training. Weightlifting may influence fascia remodeling but may not significantly change fascia thickness [15,29,35]. However, a more definitive conclusion could be drawn from a direct comparison study. Importantly, we did not detect a notable difference in TLF thickness between the LBP and non-LBP weightlifters. It aligned with the study on amateur athletes [13] but differed from the study by Pirri et al. on the general population [12]. This further suggests that in the context of long-term high-weight loading, morphological change such as thickness may not be a sufficient indicator of fascia health. Instead, mechanical properties such as stiffness may be more closely related to the adaptive remodeling and dynamic role of TLF. Future research should consider stiffness or dynamic evaluation of gliding to gain deeper insights into fascia properties.

The increased thickness in the LBP population in Pirri et al's study could be explained by the adaption of TLF [12,43–45]. However, weightlifters with LBP experience TLF remodeling from both local pathological conditions and routine high-loading training. The interaction of these two adaptations may counteract thickness changes in the LBP group or elevate thickness in the non-LBP group due to daily training. Additionally, racial variation could be a possible factor affecting fascia thickness or stiffness. Our study was conducted on an Asian population, whereas Pirri et al's study focused on Caucasians. A comparative study with matched race, loading, and LBP history may provide more conclusive results.

### Side-to-side differences in thickness and stiffness

Chronic LBP was found as a factor affecting between-side differences in stiffness but not thickness. This further suggests that stiffness may be a more informative indicator of fascia health than thickness. When comparing both sides, the thickness of the non-dominant side was higher than that of the dominant side. This asymmetry likely reflects sport-specific mechanical adaptations rather than pain-related alterations. During weightlifting, the non-dominant side often remains in a relatively shortened position, whereas the dominant side undergoes greater lengthening strain. These asymmetric loading patterns may lead to localized fascial remodeling and increased thickness on the non-dominant side, independent of LBP. The TLF is composed of multiple layers of collagen fibers oriented according to the principal direction of force transmission [35], allowing adaptive remodeling under long-term, repetitive loading. Therefore, the observed side-to-side difference in TLF thickness appears to represent an adaptation to chronic, asymmetric loading patterns typical of weightlifting movements.

In contrast, no side-to-side discrepancy was detected in stiffness. This may suggest that, unlike thickness adaption, the stiffness may not be notably different in response to loading patterns (such as shortening or lengthening) but could be more sensitive to disorganization or pathological conditions such as chronic inflammation. Although the side-to-side difference in stiffness did not reach statistical significance, possibly due to the limited sample size, the patterns between the LBP and non-LBP groups were distinct. In the non-LBP group, stiffness was higher on the non-dominant side, whereas in the LBP group, stiffness was higher on the dominant side. The presence of LBP may alter the side-to-side stiffness difference. Future studies with larger sample sizes may provide more definitive information.

### Limitations

The cross-sectional design limits causal inferences between TLF stiffness and chronic LBP. While the stiffness cohort was relatively small, the observed effect sizes were substantial. Additionally, the findings may be specific to weightlifters due to their unique spinal loading patterns, potentially limiting generalizability to other athletic populations. Furthermore, psychosocial factors were not assessed. Although such factors may interact with pain perception and chronicity, the present study primarily focused on the biomechanical characteristics of the thoracolumbar fascia. Future studies integrating both biomechanical and psychosocial dimensions would provide a more comprehensive understanding of LBP mechanisms in athletes.

### Conclusion

Elite weightlifters with chronic LBP exhibited significantly higher TLF stiffness (42%) without accompanying thickness changes, suggesting stiffness is a more sensitive indicator of TLF adaptation. These findings highlight the potential clinical value of assessing and managing TLF stiffness in prevention and rehabilitation programs. Future longitudinal studies are warranted to establish causality and develop targeted interventions.

> Key points
>
> - Elite weightlifters with chronic low back pain (LBP) showed 42% higher thoracolumbar fascia (TLF) stiffness but no difference in thickness compared to pain-free athletes.
>
> - TLF stiffness, not thickness, may serve as a more sensitive biomarker for LBP, suggesting its utility in assessing TLF health.
>
> - Rehabilitation programs for weightlifters should prioritize managing TLF stiffness to mitigate LBP and enhance performance longevity.

## Supporting information

**S1 Table. Intra-rater reliability of TLF measurements using ultrasound.**
(DOCX)

**S1 Checklist. Inclusivity in global research questionnaire.**
(DOCX)

**S1 File. STROBE checklist.**
(DOCX)

## Acknowledgments

The authors would like to acknowledge the support of Guangdong Sports Training Center and acknowledge, Prof. CS's Prof. SNF's research team members for their research assistance and feedback.

## Author contributions

**Conceptualization:** Emma Feng Ming Zhou, Siu Ngor Fu, Carla Stecco.

**Data curation:** Carmelo Pirri, Xiaoxiao Zhao, Juhua Peng, Tao Wen, Jianhui Fang.

**Formal analysis:** Emma Feng Ming Zhou, Xiaoxiao Zhao, Siu Ngor Fu, Carla Stecco.

**Investigation:** Emma Feng Ming Zhou, Juhua Peng, Chufeng Zhou.

**Methodology:** Emma Feng Ming Zhou, Carmelo Pirri, Siu Ngor Fu, Carla Stecco.

**Resources:** Juhua Peng, Tao Wen, Jianhui Fang, Chufeng Zhou, Siu Ngor Fu, Carla Stecco.

**Software:** Carmelo Pirri.

**Supervision:** Siu Ngor Fu, Carla Stecco.

**Validation:** Emma Feng Ming Zhou, Carmelo Pirri.

**Visualization:** Emma Feng Ming Zhou.

**Writing – original draft:** Emma Feng Ming Zhou.

**Writing – review & editing:** Emma Feng Ming Zhou, Siu Ngor Fu, Carla Stecco.

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
