## [Decision Letter · Decision Letter 0]

2 Sep 2025

Dear Dr. Fu,

Thank you for submitting your manuscript to PLOS ONE. After careful consideration, we feel that it has merit but does not fully meet PLOS ONE’s publication criteria as it currently stands. In particular, one reviewer raised substantial methodological concerns, particularly regarding the definition of chronic low back pain, the lack of measures to clearly establish the link between fascia and pain, and the absence of key variables that could influence interpretation, while the others were more positive. Although the reviewers expressed diverging opinions, we would like to invite you to submit a revised version of the manuscript that addresses the points raised during the review process, modifying the manuscript accordingly. Please note that the journal reserves the right to make a final decision on the basis of the revised submission.

We look forward to receiving your revised manuscript.

Kind regards,

Alice Berardo

Academic Editor

PLOS ONE

3. In the online submission form, you indicated that [The datasets supporting the conclusions of this article are available from the corresponding author upon reasonable request].

Additional Editor Comments:

Reviewer #1:

Reviewer #2:

Reviewer #3:

Reviewers' comments:

Reviewer's Responses to Questions

**Comments to the Author**

1. Is the manuscript technically sound, and do the data support the conclusions?

Reviewer #1: Yes

Reviewer #2: Yes

Reviewer #3: No

2. Has the statistical analysis been performed appropriately and rigorously?

Reviewer #1: Yes

Reviewer #2: I Don't Know

Reviewer #3: No

3. Have the authors made all data underlying the findings in their manuscript fully available?

Reviewer #1: Yes

Reviewer #2: Yes

Reviewer #3: No

4. Is the manuscript presented in an intelligible fashion and written in standard English?

Reviewer #1: Yes

Reviewer #2: Yes

Reviewer #3: Yes

Reviewer #1: Thank you for the opportunity to review your manuscript titled “Comparative Study of Thoracolumbar Fascia Changes in Weightlifters with and without Low Back Pain.” I would like to commend you and your team for the quality and relevance of this work. Your study addresses an important topic in sports medicine and rehabilitation, and the methodological rigor is evident throughout.

Strengths of the manuscript:

The use of shear wave elastography (SWE) to assess thoracolumbar fascia (TLF) stiffness is both innovative and well-justified.

The methodology is clearly described, and intra-rater reliability is robust.

Statistical analyses are appropriate and account for relevant covariates.

The findings are clinically meaningful, suggesting that stiffness may be a more sensitive biomarker than thickness for chronic low back pain (LBP) in elite athletes.

Suggestions for improvement:

Data Availability: While the statement that data are available upon reasonable request is acceptable, I encourage you to consider depositing anonymized datasets in a public repository to align with PLOS ONE’s open data policy.

Sample Size: The stiffness cohort is relatively small (n=17), which may limit generalizability. Nonetheless, the effect sizes are substantial and well-reported.

Language and Clarity: The manuscript is well-written and intelligible. Minor grammatical edits could further improve readability.

Figures and Tables: The figures are informative and well-labeled. Enhancing their resolution would benefit the final publication.

Ethics and Transparency: Ethical approval and informed consent are clearly stated. The absence of competing interests and funding bias is appreciated.

Conclusion: Your study provides valuable insights into the biomechanical adaptations of fascia in elite athletes and supports the clinical relevance of stiffness as a target in LBP management. I recommend acceptance after minor revisions, particularly regarding data availability.

Reviewer #2: THANKS FOR A CLEAR REPORT ON AN INTERESTING SUBJECT

The DISCUSSION needs some elaboration on the following results:

- 42% higher TLF stiffness in elite weightlifters with LBP than those without.

- This increase in stiffness was noted on the dominant side but not on the non-dominant side.

- LBP was found to be a factor influencing the side-to-side differences in stiffness

WHAT CAUSES STIFFNESS IN FASCIA? DOES PROTECTIVE MUSCLE SPASM ON THE DOMINANT SIDE NOT PLAY A ROLE?

WILL THIS STIFFNESS DISAPPEAR IF LBP DISAPPEARS? - AS THE PARTICIPANTS WITHOUT LBP FOR 3 MONTHS WERE USED?

There were no significant differences in TLF thickness between the two groups.

TLF on the non-dominant side was thicker than that on the dominant side

LBP was not found to be a factor influencing the side-to-side differences in thickness

WHAT WOULD CAUSE THIS THICKNESS ON THE DOMINANT SIDE - EVEN IN PEOPLE WITHOUT LBP

Reviewer #3: This study, which used ultrasound measurements to compare the thickness and stiffness of fascia in weightlifters with and without chronic low back pain (CLBP), aimed to discuss the characteristics of fascia in those with CLBP.

Significant problems exist throughout the paper. First, the pathophysiology of CLBP is not accurately captured, and the methodology to prove that the pain is fascial-derived nociceptive pain is lacking. Unfortunately, this paper requires a methodological overhaul.

# Major Issues

1. While this study investigates CLBP in weightlifters, the definition of low back pain itself is ambiguous and inadequate. Defining chronic pain in athletes is often difficult; for example, it must be clarified whether the pain occurs during daily life or only during competition.

2. The paper investigates the relationship between fascia and pain. Therefore, it should have at least examined mechanical pain thresholds on the fascia (nociceptors). As noted above, without knowing the nature of the pain and lacking information on spinal degenerative findings, it is overly simplistic to conclude a relationship between vague pain and the fascia. Please review the relevant paper [PMID: 34640360].

3. When vaguely defining "chronic low back pain," psychosocial factors must always be measured simultaneously and considered as variables.

---

# Minor Issues

1. While exclusion criteria were set for certain conditions, it is not documented whether X-rays or MRI scans were actually performed on all subjects. The rationale for this is not documented, and if MRI scans were performed, variables such as the degree of degeneration at each vertebral level (e.g., Pfirrman grade), as well as the presence of LDH, disc degenerations or Modic changes, should have been considered.

2. The introduction section states the study hypothesis, but no effect size estimates or sample size calculations were performed to test this hypothesis.

**Do you want your identity to be public for this peer review?** For information about this choice, including consent withdrawal, please see our Privacy Policy

Reviewer #1: No

Reviewer #2: No

Reviewer #3: **Yes: ** TATSUNORI IKEMOTO

---

## [Author Response · Author response to Decision Letter 1]

10 Nov 2025

Dear Editor,

RE: PONE-D-25-40990

Title: Comparative Study of Thoracolumbar Fascia Changes in Weightlifters with and without Low Back Pain

The authors would like to thank you and the Reviewers for all of your time and effort devoted to the review of our aforementioned manuscript. Where applicable, we have revised our manuscript as suggested by you and the Reviewers. Below are our specific replies to yours and the Reviewer’s insightful comments/concerns.

Reviewer #1 (The revisions are highlighted with green in the manuscript)

We sincerely thank the reviewer for the positive and encouraging feedback. We truly appreciate the recognition of the methodological rigor and clinical relevance of our study. All suggestions have been carefully considered and addressed as follows.

Comment 1 Data Availability: While the statement that data are available upon reasonable request is acceptable, I encourage you to consider depositing anonymized datasets in a public repository to align with PLOS ONE’s open data policy.

Response: We appreciate this helpful suggestion. The datasets generated and analyzed during the current study contain individual-level information from elite athletes. Public sharing of these data could risk participant re-identification and breach confidentiality.

In accordance with the institutional collaboration agreement between The University and Sports Training Center, all data and related materials are considered confidential and jointly owned by both institutions. The agreement explicitly prohibits data sharing with third parties without written consent from both institutions.

Therefore, the datasets are not publicly available. Anonymized data supporting the main findings are available from the corresponding author upon reasonable request and with prior approval from the institutional ethics board.

The statement has been updated accordingly:

“Anonymized data supporting the main findings are available from the corresponding author upon reasonable request and with prior approval from The Hong Kong Polytechnic University Institutional Review Board.” (Please see lines 358-360)

Comment 2 Sample Size: The stiffness cohort is relatively small (n=17), which may limit generalizability. Nonetheless, the effect sizes are substantial and well-reported.

Response:

Thank you for this insightful comment. We agree that the relatively small sample size may limit generalizability. This limitation primarily reflects practical challenges inherent in recruiting elite athletes, whose training schedules and competition commitments restrict research participation. Nonetheless, all participants met strict inclusion criteria, and effect sizes were substantial, supporting the robustness of the findings. This limitation has been acknowledged in the Limitations section.

Comment 3 Language and Clarity: The manuscript is well-written and intelligible. Minor grammatical edits could further improve readability.

Response: Thank you for this valuable comment. The entire manuscript has been carefully proofread and refined for grammar and clarity.

Comment 4 Figures and Tables: The figures are informative and well-labeled. Enhancing their resolution would benefit the final publication.

Response: We appreciate this suggestion. All figures have been replaced with higher-resolution versions to ensure clarity and visual quality.

Comment 5 Ethics and Transparency: Ethical approval and informed consent are clearly stated. The absence of competing interests and funding bias is appreciated.

Response: Thank you for acknowledging these aspects.

Comment 6 Conclusion: Your study provides valuable insights into the biomechanical adaptations of fascia in elite athletes and supports the clinical relevance of stiffness as a target in LBP management. I recommend acceptance after minor revisions, particularly regarding data availability.

Response: We sincerely appreciate the reviewer’s supportive recommendation and constructive feedback.

Reviewer #2 (The revisions are highlighted with blue in the manuscript)

We thank the reviewer for recognizing the value of our study and for the thoughtful suggestions that have helped us strengthen the Discussion.

Comment 1 What causes stiffness in fascia? Does protective muscle spasm on the dominant side not play a role?

Response: Thank you for this insightful comment.

We agree that the causes of fascial stiffness are multifactorial. We have revised the Discussion section to provide a more comprehensive explanation of the possible mechanisms contributing to increased fascial stiffness.

“Fascial stiffness may arise from multiple interacting mechanisms. Chronic mechanical overload and microtrauma can trigger extracellular matrix (ECM) remodeling, characterized by increased collagen deposition, altered fiber orientation, and excessive cross-linking, which reduce elasticity and the gliding ability of tissue layers.15,28,29 In addition, chronic inflammation can promote fibrosis through cytokine-mediated activation of fibroblasts and myofibroblasts, while hyaluronan accumulation increases connective tissue viscosity and adhesion between fascial layers, further elevating passive stiffness.33-37 These structural and biochemical changes impair the tissue’s ability to deform and transmit loads efficiently, predisposing the fascia to become stiffer and less resilient.” (Please see lines 216-224)

Regarding protective muscle spasm, we agree that it can transiently increase tissue stiffness; however, it is generally an acute, short-lived response to nociceptive input and typically resolves as pain subsides. In the current study, SWE measurements were performed under standardized resting conditions, minimizing active muscle tension. Therefore, the elevated TLF stiffness observed is unlikely to be attributed to transient protective spasm.

Nonetheless, we acknowledge that chronic pain may induce sustained low-level muscle activation, which can impose continuous mechanical tension on the fascia and, over time, drive extracellular matrix remodeling and collagen cross-linking. This process may secondarily increase passive fascial stiffness even in a relaxed state. We have clarified this interpretation in the Discussion section:

“Conversely, persistent or recurrent LBP itself may further exacerbate fascial stiffness. Chronic pain has been shown to induce sustained low-level muscle activation and altered motor control,40,41 which impose continuous mechanical tension on the fascia and facilitate extracellular matrix remodeling over time.14” (Please see lines 243–246)

Comment 2 Will this stiffness disappear if LBP disappears? - As the participants without lbp for 3 months were used?

Response: We appreciate this insightful question. We agree that the temporal relationship between pain remission and fascial stiffness recovery remains unclear. In our study, weightlifters in the non-LBP group were pain-free for at least three months, consistent with previous definitions of chronic LBP remission. This ensured that recent pain episodes were unlikely to influence their neuromuscular status.

However, residual fascial or neuromuscular adaptations may persist even after symptom resolution, as structural remodeling (e.g., collagen cross-linking or fibrosis) often reverses more slowly than pain symptoms. This has been noted in the Discussion as a consideration

“Furthermore, it remains unclear whether fascial stiffness fully normalizes once LBP resolves. Although participants in the non-LBP group were pain-free for at least three months,20-22 residual fascial or neuromuscular adaptations from prior loading or pain episodes may persist. Structural remodeling, such as collagen cross-linking or fibrosis, may reverse more slowly than pain symptoms, potentially leaving subtle mechanical alterations even after symptom resolution. Longitudinal or interventional studies tracking stiffness changes before and after recovery would help clarify the reversibility and temporal dynamics of these adaptations.” (Please see lines 253–259)

Comment 3 There were no significant differences in TLF thickness between the two groups.

TLF on the non-dominant side was thicker than that on the dominant side

LBP was not found to be a factor influencing the side-to-side differences in thickness

WHAT WOULD CAUSE THIS THICKNESS ON THE DOMINANT SIDE - EVEN IN PEOPLE WITHOUT LBP

Response: Thank you for this insightful comment. As clarified in the revised Discussion (Side-to-side differences in thickness and stiffness section), the thickness asymmetry between the dominant and non-dominant sides likely reflects sport-specific mechanical adaptations rather than pain-related alterations.

“This asymmetry likely reflects sport-specific mechanical adaptations rather than pain-related alterations. During weightlifting, the non-dominant side often remains in a relatively shortened position, whereas the dominant side undergoes greater lengthening strain. These asymmetric loading patterns may lead to localized fascial remodeling and increased thickness on the non-dominant side, independent of LBP. The TLF is composed of multiple layers of collagen fibers oriented according to the principal direction of force transmission,29 allowing adaptive remodeling under long-term, repetitive loading. Therefore, the observed side-to-side difference in TLF thickness appears to represent an adaptation to chronic, asymmetric loading patterns typical of weightlifting movements.”(Please see lines 300–308).

Reviewer #3 (The revisions are highlighted with yellow in the manuscript)

We thank the reviewer for the careful and detailed review. We understand the concerns regarding methodology and definitions and have revised the manuscript accordingly to clarify these issues.

Comment 1 While this study investigates CLBP in weightlifters, the definition of low back pain itself is ambiguous and inadequate. Defining chronic pain in athletes is often difficult; for example, it must be clarified whether the pain occurs during daily life or only during competition.

Response: We sincerely thank the reviewer for this thoughtful and important comment. We fully agree that the definition of chronic LBP in athletes should clearly specify the contexts in which pain occurs. In our study, data were indeed collected based on cases where pain was present during both daily life and athletic training, persisting or fluctuating for more than 3 months, with an average NPRS score ≥3 over the past 7 days. However, this detail was not explicitly stated in the previous version.

To address this, we have revised the definition in the Methods – Participant Recruitment section to clarify this point. The revised wording aligns with widely accepted definitions of chronic LBP in both athletes and the general population, ensuring conceptual and methodological consistency.

“...chronic LBP characterized by non-specific pain or discomfort between the lower rib margins and the buttock creases, present during both daily life and athletic training, persisting or fluctuating for more than 3 months,20-22 with an average pain intensity of at least 3 out of 10 on the Numeric Pain Rating Scale (NPRS) over the past 7 days.23” (Please see line 91-94)

Comment 2 The paper investigates the relationship between fascia and pain. Therefore, it should have at least examined mechanical pain thresholds on the fascia (nociceptors). As noted above, without knowing the nature of the pain and lacking information on spinal degenerative findings, it is overly simplistic to conclude a relationship between vague pain and the fascia. Please review the relevant paper [PMID: 34640360].

Response: We sincerely thank the reviewer for this insightful and valuable comment, as well as for directing us to the relevant reference (PMID: 34640360). We fully agree that nociceptive sensitivity represents an essential aspect of pain mechanisms. In the present study, our focus was limited to the structural and mechanical characteristics (thickness and shear modulus) of the thoracolumbar fascia, rather than nociceptive function or the identification of specific pain sources. Therefore, assessments such as mechanical pain thresholds were not included, as our aim was to characterize fascia morphology and material properties in athletes with and without LBP.

In response to this helpful suggestion, we have added a statement in the Discussion acknowledging that the underlying nociceptive mechanisms were not examined, and that future studies incorporating both mechanical and nociceptive assessments are warranted to clarify this relationship:

“While this study focused on the structural and mechanical characteristics of the thoracolumbar fascia, the underlying nociceptive mechanisms were not examined. Given that fascia is richly innervated and may contribute to pain sensitization when its mechanical properties are altered,41 future studies incorporating both mechanical and nociceptive assessments are warranted to clarify this relationship.” (Please see lines 260-264)

Comment 3 When vaguely defining "chronic low back pain," psychosocial factors must always be measured simultaneously and considered as variables.

Response: We sincerely appreciate this thoughtful and important comment. We fully agree that psychosocial factors may interact with pain perception and play a role in the chronicity of LBP. In the present study, our focus was limited to the biomechanical characteristics of the thoracolumbar fascia in elite athletes, and psychosocial aspects were not included in the analysis. We have now acknowledged this as a study limitation and noted that future work combining biomechanical and psychosocial assessments would provide a more comprehensive understanding of LBP mechanisms:

“Furthermore, psychosocial factors were not assessed. Although such factors may interact with pain perception and chronicity, the present study primarily focused on the biomechanical characteristics of the thoracolumbar fascia. Future studies integrating both biomechanical and psychosocial dimensions would provide a more comprehensive understanding of LBP mechanisms in athletes.” (Please see lines 322-326)

Comment 4 While exclusion criteria were set for certain conditions, it is not documented whether X-rays or MRI scans were actually performed on all subjects. The rationale for this is not documented, and if MRI scans were performed, variables such as the degree of degeneration at each vertebral level (e.g., Pfirrman grade), as well as the presence of LDH, disc degenerations or Modic changes, should have been considered.

Response: We sincerely thank the reviewer for this thoughtful and important comment. We fully agree that imaging examinations such as MRI or X-rays can provide valuable information regarding spinal pathology. However, in the present study, such imaging assessments were not performed for all participants, as this would not be feasible or ethically justified for elite athletes without clinical indications.

Instead, participant eligibility was confirmed through a review of existing medical records and evaluations by team physicians to exclude pathological conditions. This approach is consistent with previous similar studies.

To clarify this point, we have added a statement in the Methods – Participant Recruitment section:

“These conditions were excluded based on medical records and evaluations conducted by team physicians.” (Please see lines 98-99)

Comment 5 The introduction section states the study hypothesis, but no effect size estimates or sample size calculations were performed to test this hypothesis.

Response: We sincerely thank the reviewer for this valuable comment. We acknowledge that an a priori sample size calculation was not conducted, mainly due to the limited availability of elite athletes. However, the statistical power of the main ANCOVA analyses was calculated and reported to verify that the sample size was sufficient to detect meaningful between-group differences. This clarifying statement has been added in the Statistical Analysis section

“In addition, the statistical power of the main ANCOVA analyses was calculated and reported to verify that the sample size was sufficient to detect meaningful betwee

---

## [Decision Letter · Decision Letter 1]

26 Nov 2025

Comparative study of thoracolumbar fascia changes in weightlifters with and without low back pain

PONE-D-25-40990R1

Dear Dr. Fu,

We’re pleased to inform you that your manuscript has been judged scientifically suitable for publication and will be formally accepted for publication once it meets all outstanding technical requirements.

Kind regards,

Alice Berardo

Academic Editor

PLOS ONE

Additional Editor Comments (optional):

Since only one reviewer accepted to review the revised manuscript, I revised it personally, and I confirmed that all the highlighted comments have been addressed.

Reviewers' comments:

Reviewer's Responses to Questions

**Comments to the Author**

Reviewer #2: All comments have been addressed

2. Is the manuscript technically sound, and do the data support the conclusions?

Reviewer #2: Yes

3. Has the statistical analysis been performed appropriately and rigorously?

Reviewer #2: Yes

4. Have the authors made all data underlying the findings in their manuscript fully available?

Reviewer #2: Yes

5. Is the manuscript presented in an intelligible fashion and written in standard English?

Reviewer #2: Yes

Reviewer #2: Thanks for addressing the reviewers' comments adequately. The report now represents your thoughts on the results of your study

**Do you want your identity to be public for this peer review?** For information about this choice, including consent withdrawal, please see our Privacy Policy

Reviewer #2: **Yes: ** Dr Ina Diener. Physiotherapy Clinician and Lecturer

---

## [Editor Report · Acceptance letter]

PONE-D-25-40990R1

PLOS One

Dear Dr. Fu,

I'm pleased to inform you that your manuscript has been deemed suitable for publication in PLOS One. Congratulations! Your manuscript is now being handed over to our production team.

Kind regards,

on behalf of

Dr. Alice Berardo

Academic Editor

PLOS One